# The Conditioning of Adjuvant Chemotherapy for Stage II and III Rectal Cancer Determined by Postoperative Pathological Characteristics in Romania

**DOI:** 10.3390/medicina59071224

**Published:** 2023-06-29

**Authors:** Horia-Dan Liscu, Bogdan-Radu Liscu, Ruxandra Mitre, Ioana-Valentina Anghel, Ionut-Lucian Antone-Iordache, Andrei Balan, Simona Coniac, Andreea-Iuliana Miron, Georgian Halcu

**Affiliations:** 1Discipline of Oncological Radiotherapy and Medical Imaging, University of Medicine and Pharmacy “Carol Davila”, 020021 Bucharest, Romania; horia-dan.liscu@drd.umfcd.ro (H.-D.L.);; 2Radiotherapy Department, Colțea Clinical Hospital, 030167 Bucharest, Romania; 3Medical Oncology Department, Colțea Clinical Hospital, 030167 Bucharest, Romania; 4Discipline of Pathological Anatomy, University of Medicine and Pharmacy “Carol Davila”, 020021 Bucharest, Romania

**Keywords:** adjuvant chemotherapy, neoadjuvant radiotherapy, neoadjuvant radiochemotherapy, rectal cancer, tumor downstage, Romanian oncologists

## Abstract

The management of locally advanced rectal cancer (LARC) suffered changes thanks to the development of improved surgical procedures, radiation delivery, and chemotherapy. Although treatment options improved individually, the optimal order is still debated. Neoadjuvant chemo-radiotherapy followed by total mesorectal excision (TME) has been the “golden standard” for locally advanced rectal cancer. There is no common ground in international guidelines on the indications of adjuvant chemotherapy (ADJCHT), with differences between the American, European, and Japanese guidelines. This paper studies the preferences of Romanian oncologists in prescribing ADJCHT. We conducted a single-institution, retrospective study of all nonmetastatic, ECOG 0-1 LARC patients staged II-III who underwent TME and were admitted to the Oncology or Radiotherapy Department of Colțea Clinical Hospital, Bucharest between January 2017 and March 2021. A total of 186 patients were included in the study. A positive correlation was found between ADJCHT and each of the following: (y)pT > 2, (y)pN > 0, and the presence of perineural invasion (PNI+). A strong positive correlation was found between ADJCHT and the presence of at least one risk factor: (y)pT > 2, (y)pN > 0, PNI+, lymphovascular invasion, positive margins, or tumor grade > 1. Tumor downstaging decreased the risk of metastases in the first 2 years and was associated with the use of neoadjuvant radiotherapy, while adding neoadjuvant chemotherapy increased the chance of nodal downstaging. ADJCHT practice for LARC in Romania follows either NCCN or ESMO guidelines, at the discretion of the oncologist, due to the lack of national guideline.

## 1. Introduction

### 1.1. Overview

Locally advanced rectal cancer is most often treated with the three pillars of oncology: surgery, radiotherapy, and chemotherapy, with the classic order being neoadjuvant radiochemotherapy, followed by surgery, and finally by systemic therapy. Adjuvant chemotherapy (ADJCHT) after neoadjuvant treatment in rectal cancer is recommended by the American National Comprehensive Cancer Network [1], European Society for Medical Oncology [2], and Japanese Society for Cancer of the Colon and Rectum [3] guidelines on a routine basis for patients who have been treated with standard neoadjuvant treatment short-course radiotherapy (SCRT) or long-course radiochemotherapy (LCRCT) for stage III. In the case of stage II rectal cancer, however, there are differences between the NCCN guideline, which recommends the use of adjuvant chemotherapy systematically, versus the ESMO and JSCCR guidelines, which recommend the use of adjuvant chemotherapy only in the presence of an increased risk of relapse [4].

In Romania, there is no national guideline for the treatment of colorectal cancer, with each oncologist having the possibility to follow international protocols and use free judgment in the decision to administer adjuvant chemotherapy. In 2021, a political initiative to promote a national cancer plan [5] emerged, which aims to improve the management of oncological diseases. A guideline was proposed that monitors the colorectal cancer patient pathway and recommends the use of adjuvant chemotherapy for all stage II and III rectal cancers, rather than mimicking the NCCN treatment guideline, without presenting its own nationally conducted studies.

This study aims to identify the criteria chosen by Romanian oncologists to administer adjuvant chemotherapy in patients with locally advanced rectal cancer, in the absence of a national guideline that clearly indicates this. The secondary objective is to analyze the relationships between the use of neoadjuvant treatment, the rate of tumor or nodal downstage, and the rate of recurrence or metastasis at 2 years.

### 1.2. Background

The oncological treatment of locally advanced rectal cancer has undergone many changes in recent decades, with it now using all three major oncological therapeutic options: radiotherapy, chemotherapy, and surgery. However, colorectal cancer still remained the second leading cause of mortality globally in 2020 [6] and has an increasing incidence in Central and Eastern Europe, including Romania, where more than 4800 new cases of rectal cancer were diagnosed in 2020 [7]. The appropriate administration order of the therapeutic methods was studied at the end of the 20th century, adding radiotherapy and chemotherapy to the neoadjuvant treatment scheme and establishing the total mesorectal excision (TME) technique as the gold-standard surgical procedure [8,9]. The 5-year local recurrence rate ranged from 15 to 40% [10,11] in the 1980s, decreasing to 4–15% in recent years with technological advances and the optimization of cancer treatment [12,13,14]. However, the reported metastasis rate (MR) for stage II-III rectal cancer is as high as 25%, with a slight improvement after the introduction of neoadjuvant radiosensitizing chemotherapy [15,16]. 

In recent years, the possibility of total neoadjuvant treatment (TNT) including short-course radiotherapy or long-course radiochemotherapy, given before or after 3–4 months of fluoropyrimidine and oxaliplatin chemotherapy, has been studied, with superior results for achieving a complete pathological response (pCR) and better disease-free survival (DFS) and overall survival (OS) [17,18]. The RAPIDO trial [19] shows an improvement in the rate of local recurrence and metastasis rate, and the OPRA trial [20] even proposes the “watch and wait” organ preservation strategy and omitting surgery in patients with complete response [21]. 

The total neoadjuvant treatment strategy is not yet recommended for all stages of locally advanced rectal neoplasm (LARC) [1] and is not adopted in all centers, which is why standard neoadjuvant treatment using short-course radiotherapy or long-course radiochemotherapy remains the basic indication in locally advanced rectal cancer. 

In the case of adjuvant chemotherapy, the EORTC 22921 study indicated, at the first analysis, a benefit of adjuvant chemotherapy for local control [22], but at the 10-year analysis, it actually showed no survival benefit [23]. Another reference study is the Italian study published in 2014 showing no benefit of adjuvant fluoropyrimidine monotherapy for overall survival [24]. A large meta-analysis comprised of four studies concluded that adjuvant chemotherapy does not benefit overall survival, disease-free survival, or metastasis rate, except for upper rectal cancers [25]. The ADORE study also added oxaliplatin to the adjuvant treatment regimen and obtained better results for disease-free survival at 6 years (68% vs. 57% HR 0.63, 95% CI 0.43–0.93), with no difference in survival [26,27]. 

## 2. Materials and Methods

### 2.1. Patients and Design

We conducted a retrospective study on 186 patients with clinically or pathologically diagnosed locally advanced rectal cancer in stages II or III who presented to the Radiotherapy or Medical Oncology Department of Colțea Clinical Hospital, Bucharest between January 2017 and March 2021. All patients included in the study underwent surgery using the TME technique via the low anterior or abdominoperineal approach and had an ECOG performance status of 0 or 1 at the time of surgery. 

Epidemiological information was collected on gender, age, and tumor location described by the surgeon during colonoscopy at the lower rectal, middle rectal, upper rectal, or rectosigmoid junction. All patients were clinically and pathologically staged, and the following histopathological report data were analyzed: tumor and nodal stage (pT and pN), resection margins (Postop), histopathological grade (Grade), and the presence/absence of lymphovascular (LVI) and perineural invasion (PNI). The extent of extramural invasion was not analyzed as not all preoperative MRIs provided this information. The tumor marker CEA was not routinely recorded in all patients and was excluded from the statistical analysis. 

### 2.2. Treatment

Patients with locally advanced rectal cancer who received neoadjuvant treatment underwent long-course external pelvic radiotherapy with doses between 45 and 50.4 Gy, using the 3D or IMRT techniques +/− chemotherapy with capecitabine at a dose of 825 mg/m^2^ twice daily during radiotherapy. TME surgery was performed for all patients 8–12 weeks after neoadjuvant treatment. 

In the case of upper rectal cancer with sigmoid extension, some patients were treated with surgery per primam, followed by adjuvant treatment according to the postoperative pathological report. 

Patients who received adjuvant chemotherapy received between 6 and 8 cycles (of 3 weeks each) of 1000 mg/m^2^ capecitabine twice daily on days 1–14, concurrently with 130 mg/m^2^ oxaliplatin on day 1, or 1250 mg/m^2^ capecitabine monotherapy twice daily on days 1–14 if biological constants did not allow platinum administration.

### 2.3. Follow-Up

Patients were followed up by the attending medical oncologist or radiation oncologist every 3 months and CT or MRI imaging investigations were performed for locoregional evaluation, as per the clinic protocol. Follow-up data from the first 2 years were analyzed for all patients and any local recurrences or distant metastasis were recorded. 

### 2.4. Statistical Analysis

Statistical analysis was performed using IBM SPSS Statistics v29 software, applying Pearson correlations and t-test statistical significance tests between the use of adjuvant chemotherapy and each of the histopathological criteria: (y)pT, (y)pN, LVI, PNI, Postop, and Grade. The same correlation algorithm was applied to identify the link between the use of neoadjuvant radiotherapy (NEORT) or radiochemotherapy (NEOCHTRT), the occurrence of local recurrence (relapse2y) or metastatic disease (metastasis2y) in the first 2 years, and the occurrence of tumor (T.downstage) and nodal (N.downstage) downstaging.

## 3. Results

Of the total 186 patients included in the study, 53.2% of patients (n = 99) were male (B) and 46.8% (n = 87) were female (F) (Table 1; Figure 1). 

The mean age of the patients was 67.42 years with a standard deviation of 11.07 years. A majority distribution of patients was observed in the 60–80 years age range (Figure 2). 

The location of the tumor was at the lower rectal (RI) level in 29.25% of cases (n = 43), at the middle rectal (RM) level in 40.14% of cases (n = 59), at the upper rectal (RS) level in 26.53% of cases (n = 49), and at the rectosigmoid junction (JONC) in 30.61% of cases (n = 45). Most of the patients had tumors located in the middle rectum (Table 2, Figure 3). 

Compared to other rectal cancer studies, our study included patients with rectal tumors located higher up at the rectosigmoid junction, as some of them received neoadjuvant long-course radiochemotherapy treatment similar to standard rectal cancer treatment. 

Pretreatment clinical staging includes 6.01% stage I patients (n = 11), 25.14% stage II patients (n = 46) and 68.85% stage III patients (n = 126) (Figure 4). 

When it comes to clinical substages, most stage II patients were found in substage IIA (rectal tumor penetrating the muscularis propria and invading the colorectal tissues, without clinically detectable adenopathy). Of the stage III patients, most could be clinically substaged in IIIB (rectal tumor invading or penetrating its own muscle, with the presence of up to six clinically detectable malign lymph nodes).

Pathological staging performed after surgery was composed of 2.69% stage 0 patients with pathological complete response after neoadjuvant treatment (n = 5), 9.68% stage I patients (n = 18), 34.95% stage II patients (n = 65), and 52.69% stage III patients (n = 98) (Figure 5). 

The substaging of patients in pathological stage II showed a clear majority of over 30% of patients in stage IIA. For stage III patients, most were pathologically substage IIIB, but with a significant 10% patients in stage IIIC.

We analyzed the correlation between adjuvant chemotherapy treatment and histopathology bulletin results by using the Pearson correlation coefficient and applying the two-tailed significance *t*-test. The correlation between adjuvant chemotherapy and a pathological T staging greater than 2 (y)pT > 2 (Figure 6) and between adjuvant chemotherapy and a positive pathological N staging (y)pN > 0 was tested. Other risk factors from the histopathology report that were taken into account were the presence of lymphovascular invasion LVI+ (Figure 7), the presence of perineural invasion PNI+ (Figure 8), positive resection margins Postop+, or a histopathological grade greater than 1 Grade > 1.

Adjuvant chemotherapy is positively correlated with a moderate score of 0.432 for a histopathological stage (y)pT3 or (y)pT4 (Table 3). A moderate positive correlation score of 0.462 is also valid for the presence of nodal invasion (y)pN > 0. Both results are statistically significant with a score of *p* < 0.001. 

Analyzing the relationship between adjuvant chemotherapy, the presence of lymphovascular invasion, the presence of perineural invasion, the tumor grade, and the postoperative resection margins, the results show a very weak positive correlation between them, with a statistically significant *p* only in the case of the correlation between adjuvant chemotherapy and the presence of perineural invasion *p* = 0.01 (Table 4). In the case of lymphovascular invasion, positive resection margins, and histopathological grade greater than 1, the correlation with adjuvant treatment was very weak and statistical significance did not exist.

In the case of correlation analysis between the use of adjuvant chemotherapy and at least one risk factor for relapse, the Pearson score is highly positive (+0.633) with statistically significant *p* < 0.001, suggesting that in the presence of at least one risk factor for relapse, the oncologist would decide to administer adjuvant chemotherapy (Table 5, Figure 9). The risk factor may represent any of the following: histopathological stage pT3 or pT4, the presence of lymphatic invasion (pN > 0), the presence of lymphovascular invasion, the presence of perineural invasion, positive resection margins, or histopathological grade 2 or 3.

Last but not least, the hypothesis that Romanian oncologists would administer adjuvant chemotherapy if neoadjuvant treatment is omitted was verified. There is a weak but statistically significant correlation supporting the omission of neoadjuvant treatment as an argument for adjuvant treatment (*p* = 0.11 for NEOCHTRT and *p* = 0.009 for NEORT) (Table 6). 

Standard neoadjuvant treatment, LCRCT, was positively correlated with the occurrence of tumor and nodal downstaging, with the latter being more significant with a correlation score of 0.436 and statistical significance at *p* < 0.001. RT alone given in neoadjuvant regimen yielded similar results for tumor downstaging, but weaker ones for nodal downstaging (Table 7, Figure 10). 

The occurrence of distant metastasis and local recurrence within the first 2 years of follow-up was recorded for the patients analyzed. Patients who received neoadjuvant treatment, either radiotherapy alone or long-course radiochemotherapy, had a nonsignificant correlation score with the metastasis rate or local recurrence rate in the first 2 years. However, paradoxically, neoadjuvant treatment was positively correlated with the local recurrence rate at 2 years and with the metastasis rate at 2 years: for neoadjuvant radiochemotherapy, local recurrence was correlated with a Pearson score of 0.131 and metastasis with a Pearson score of 0.082.

Analyzing the downstage rate, we saw that there was an inverse correlation between the tumor downstage rate after neoadjuvant treatment and the 2-year metastasis rate of −0.193, with a statistically significant *p* = 0.008 (Figure 11). Additionally, the nodal downstage rate was negatively correlated with the 2-year metastasis rate, with a Pearson score of −0.102, but this time, there was no statistical significance at *p* = 0.165. The local recurrence rate at 2 years was practically insignificant, with a correlation score close to 0 and a t-test *p* > 0.25 for both tumor downstage and nodal downstage.

## 4. Discussion

The epidemiological data analyzed in the study are similar to other studies conducted on larger cohorts in terms of mean age or sex distribution [28,29]. The present study includes a larger number of patients with upper rectal tumors, as it includes patients with tumors at the rectosigmoid junction. There is currently debate about the optimal treatment modality for high rectal cancer, with suggestions that the peritoneal limit is the key for indicating or omitting neoadjuvant treatment [30]. This principle of choosing the treatment of upper rectal cancers, located at the junction with the sigmoid, poses difficulties for surgeons and oncologists and especially depends on the correct interpretation of preoperative investigations. Sometimes, the multidisciplinary committee may consider that a high rectal cancer does not benefit from neoadjuvant treatment (considering it rather as sigmoid cancer), and then the first intervention would be surgery. In this case, depending on the stage and other postoperative molecular factors, adjuvant chemotherapy may also be omitted, similarly to rectal cancer, which does not preclude the inclusion of patients who did not receive neoadjuvant treatment in this study.

Our main objective was to analyze the association between the use of adjuvant chemotherapy and the postoperative pathological characteristics. The decision to perform adjuvant chemotherapy in patients with stage II-III rectal cancer is mainly based on the presence of a (y)pT > 2 or (y)pN > 0, or a minimum of one out of the following risk factors: a positive resection margin, the presence of lymphovascular invasion, the presence of perineural invasion, or a histopathological grade greater than 1. The only adverse factor encountered on the histopathology report that was uniquely correlated with the use of adjuvant therapy is perineural invasion, despite the fact that the Dutch PROCTOR study [31] showed that PNI+ does not necessarily predict a beneficial effect of adjuvant chemotherapy. This rationale is based more on the association between the presence of perineural invasion and inferior survival [32,33]. According to Swets et al. [32], the presence of extramural invasion, tumor budding, or perineural invasion is associated with much worse overall survival, and the presence of two out of three also decreases disease-free survival and increases the chance of distant metastases. However, none of these features, alone or all together, can predict any benefit of adjuvant chemotherapy.

In pathological stage II, there is inconsistency among oncologists, with most of them choosing to treat these patients with adjuvant chemotherapy according to the NCCN guideline [1]. However, 6.9% of pathologically stage II patients had no risk factors for relapse (n = 13) and did not receive adjuvant treatment, instead following the ESMO [2] or JSCCR [3] guidelines. This rationale is also supported by a retrospective analysis performed in 2022 in Taiwan, which supports omitting adjuvant treatment in ypT0-2N0 patients with good response to neoadjuvant treatment. Kuo et al. [34] conducted a retrospective study of 720 patients with good response after neoadjuvant treatment and found that neither overall survival nor disease-free survival is improved by adjuvant chemotherapy, even for clinically advanced patients.

The findings of other authors show that precisely the lower stages and the presence of downstaging would benefit from adjuvant chemotherapy, thus only fueling the confusion and controversy regarding this subgroup of patients. Breugom et al. [24] consider that adjuvant chemotherapy may be especially useful for patients with tumors located at an upper level, at least 10 cm from the anal margin, in terms of disease-free survival and metastasis. However, these findings do not hold true for overall survival or for tumors located at the lower or middle rectal level.

The high degree of correlation between the administration of adjuvant chemotherapy and the presence of at least one risk factor (Pearson score of +0.633) suggests that if one or more risk factors for relapse are present, a Romanian oncologist would decide to administer adjuvant chemotherapy. The regimen includes at least one antimetabolite (those classically used in Europe and the United States are 5-fluorouracil in combination with leucovorin or orally administered capecitabine, while in Asia, tegafur–gimeracil–oteracil is an option), but most experts’ recommendation is to add oxaliplatin. This combination improves disease-free survival, but the addition of oxaliplatin for overall survival is unclear. The phase II ADORE trial [26] of 321 patients with stage II or III rectal cancer treated with neoadjuvant radiochemotherapy, total mesorectal excision, and adjuvant chemotherapy with 5 FU (control arm) or FOLFOX (experimental arm) showed an overall 3-year survival difference of 85.7% versus 95.0% in favor of adjuvant oxaliplatin (HR =.456; 95% CI 0.25–0.97; *p* = 0.036). The German phase III CAO/ARO/AIO-04 trial [35] found, however, no difference between the two adjuvant groups (with or without oxaliplatin) for 3-year overall survival (88.7% vs. 88%). In our study, the decision to add oxaliplatin to the adjuvant regimen was judged for each individual patient according to biological constants and the risk of relapse. The Japanese national guideline [3] recommends the addition of oxaliplatin only in cases with a higher risk of relapse, concluding that it reduces the risk of relapse or death by about 20% compared with antimetabolite monotherapy.

Another argument for adjuvant treatment often encountered among Romanian physicians, but with less impact according to the Pearson correlation analysis, is the omission of neoadjuvant treatment; thus, there is a higher risk of locoregional or distant relapse. The American guidelines recommend both adjuvant radiochemotherapy and a course of adjuvant chemotherapy after a resection of rectal adenocarcinoma, if not previously offered [36]. Chemotherapy regimens similar to adjuvant treatment in colon cancer are used: fluoropyrimidine monotherapy (possibly administered in an accelerated Gramont-type regimen) with or without oxaliplatin and usually without the addition of irinotecan.

In terms of oncological outcome after treatment, the results obtained in this study are inferior compared to data obtained in other clinical trials [10,11,12,13,14], with a percentage of local relapse or metastasis at 2 years ranging from one quarter to one third of patients. It should be taken into account that more than 50% of the patients included in the study were stage III and not all patients received neoadjuvant treatment. The absence of an organized national population screening program in Romania, both for colorectal cancer and for other common cancers [37], is one of the causes leading to the discovery of cancer in advanced stages.

Our secondary endpoints were to analyze the correlations between the use of neoadjuvant treatment, the rate of tumor or nodal downstage, and the rate of recurrence or metastasis at 2 years. Patients who benefited from tumor downstaging also had lower odds of metastasis, confirming the good prognosis of a response to neoadjuvant treatment, as shown by the analysis in the National Cancer Database performed in the US in 2018 [38]. Local control is not associated with tumor or nodal downstaging, which contradicts a 2018 study [39]. A total of 5 patients out of 77 who received long-course radiochemotherapy had pathological complete response, this percentage of 6.5% being significantly lower than other authors’ results of 15–20% [40,41]. This may be explained by the inclusion of a large proportion of patients with pT3 stage in the study, for whom standard neoadjuvant treatment is not aggressive enough in order to obtain complete tumor regression. However, the 3-year survival of patients who had complete pathological response after neoadjuvant radiochemotherapy reported by Tan et al. [41] was 92.4%, only a few percent higher than the survival of the group that did not achieve pathological complete response (88.2%). This difference is statistically significant at *p* = 0.002, but the relatively small gap between the two groups shows that the role of total mesorectal excision performed correctly could be more important than achieving pathological complete response after neoadjuvant treatment. Thus, in patients for whom organ preservation is not established as an objective, fast neoadjuvant treatment, regimens (such as short-course radiotherapy) with immediate surgery and adjuvant chemotherapy may be an option.

### Strengths and Limitations of the Study

This study is the first study conducted on a population of patients treated in Romania with locally advanced rectal cancer that analyzes the criteria of choice of adjuvant therapy offered by oncologists. The rationale for the adjuvant treatment of rectal cancer administered in Romania is based on American, European, or Japanese guidelines.

The limitations of the study include the collection of patients from a single hospital unit (as Romania does not yet have a single national registry of cancer patients), which may have led to selection bias. Additionally, some patients were incompletely investigated (lack of CEA dosing) or showed incomplete investigation results (due to a lack of extramural invasion description on preoperative MRI scans). A retrospective analysis cannot provide strong evidence of causality; thus, we recommend future prospective analyses on this topic to confirm our findings. We included in the study patients who received neoadjuvant treatment and patients who did not receive such treatment. The primary endpoint was not influenced by neoadjuvant treatment, and the secondary endpoint analysis on downstaging was only performed on patients who received neoadjuvant treatment.

## 5. Conclusions

Patients with locally advanced rectal neoplasm treated in Romania will benefit from adjuvant chemotherapy if the patient does not have pathological stage (y)pTNM 0 or I. For pathological stage II patients who have completed neoadjuvant treatment and do not have any risk factor for recurrence, adjuvant treatment may be omitted, but this remains at the oncologist’s discretion. The appearance of tumor downstaging decreases the risk of metastases in the first 2 years and is associated with the use of neoadjuvant radiotherapy. The addition of radiosensitizing chemotherapy is associated with a higher chance of nodal downstaging.

Oncology practice in Romania runs the risk of heterogeneity between institutions, and even between physicians in the same institution, due to the lack of standardized national guidelines, large-scale national studies, and a national cancer registry that can centralize this information.

## Figures and Tables

**Figure 1 medicina-59-01224-f001:**
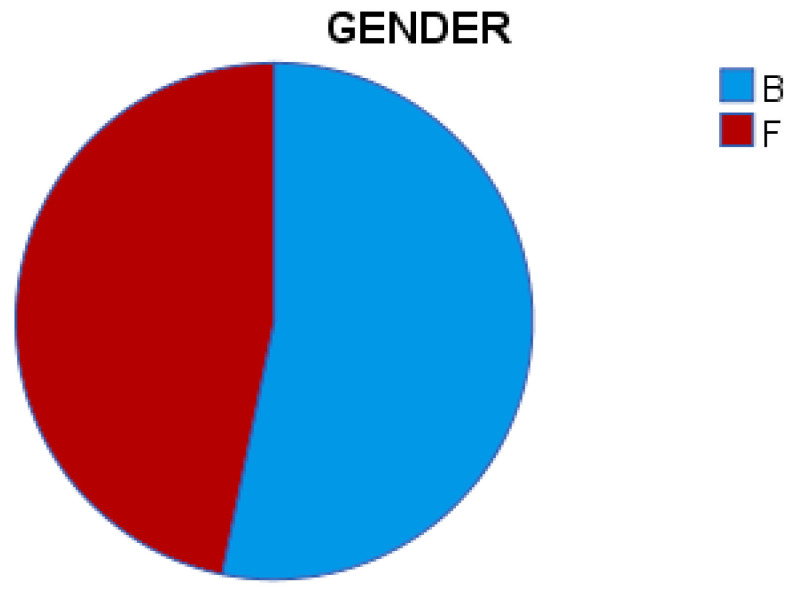
Pie chart for gender.

**Figure 2 medicina-59-01224-f002:**
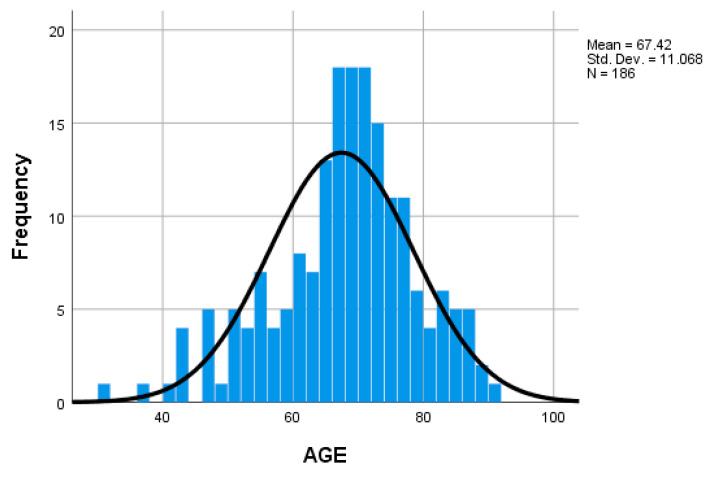
Histogram of age distribution.

**Figure 3 medicina-59-01224-f003:**
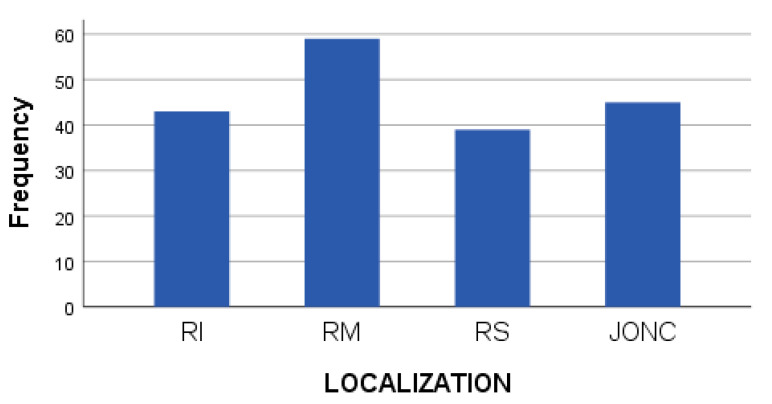
Bar chart showing the location of the tumor formation.

**Figure 4 medicina-59-01224-f004:**
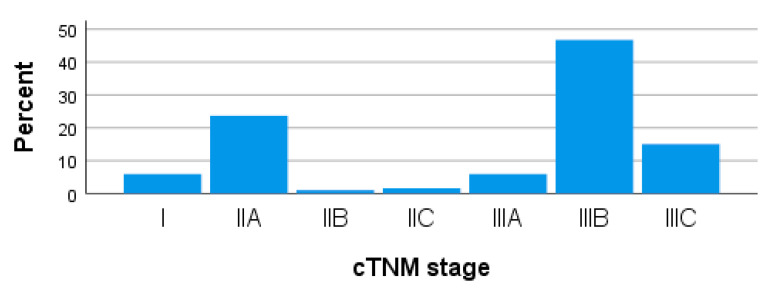
Bar chart with clinical staging cTNM of patients.

**Figure 5 medicina-59-01224-f005:**
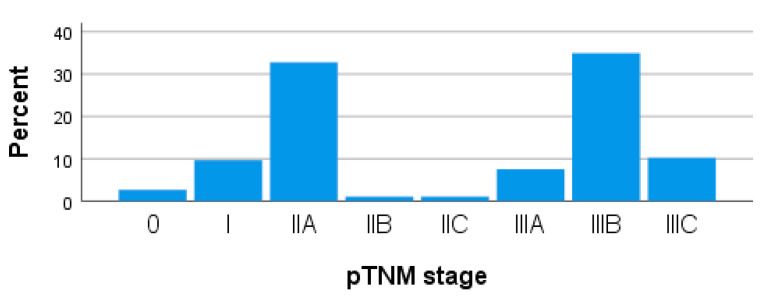
Bar chart with pathological staging pTNM of patients.

**Figure 6 medicina-59-01224-f006:**
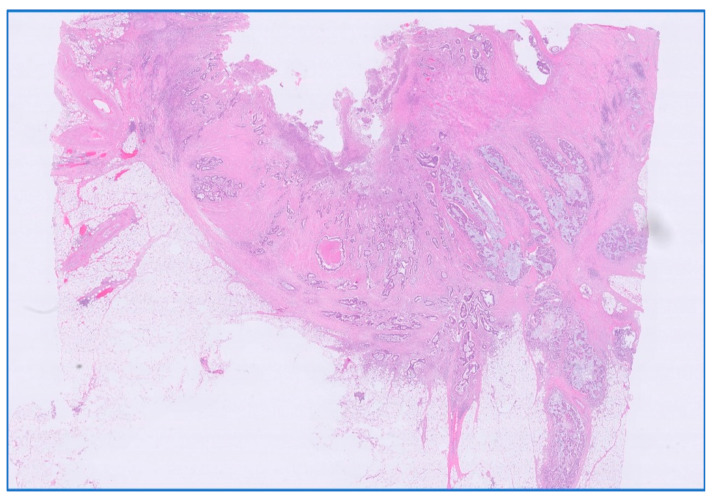
Persistent rectal adenocarcinoma after neoadjuvant radiochemotherapy; ypT3 with invasion in subserosal adipose tissue. HE stain × 0.5 magnification.

**Figure 7 medicina-59-01224-f007:**
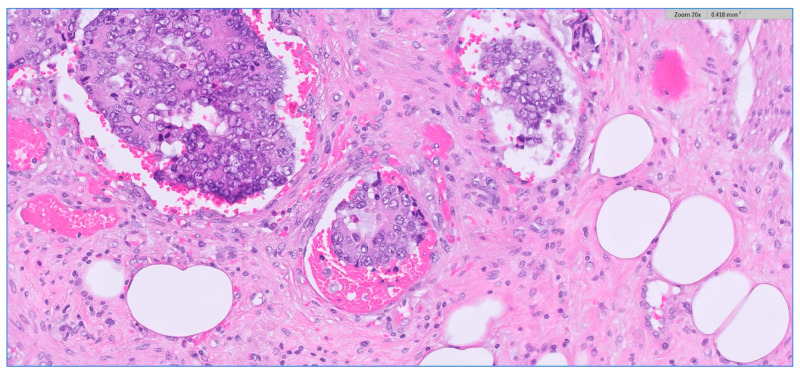
Depiction of extramural lymphovascular invasion; carcinomatous emboli of rectal adenocarcinoma. HE stain × 20 magnification.

**Figure 8 medicina-59-01224-f008:**
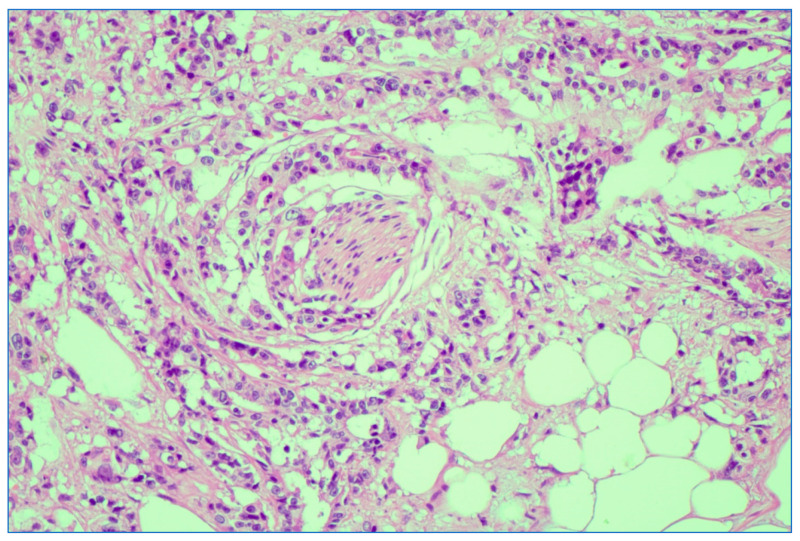
Depiction of perineural invasion; rectal adenocarcinoma. HE stain × 5 magnification.

**Figure 9 medicina-59-01224-f009:**
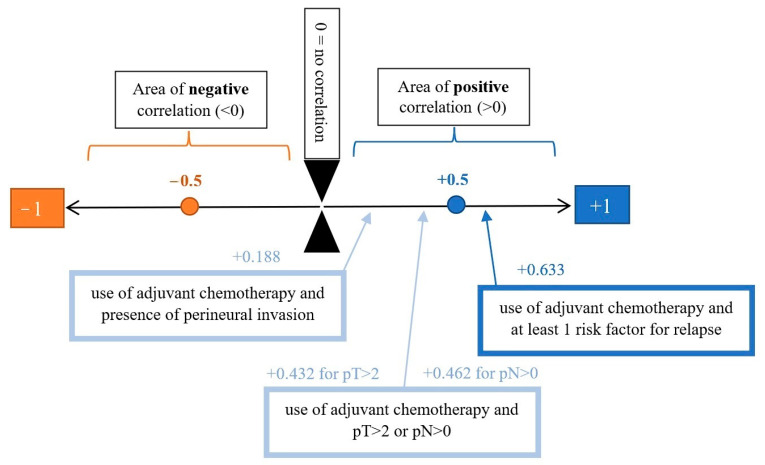
Graphical representation of the correlation between adjuvant chemotherapy and presence of perineural invasion; pT > 2 or pN > 0; minimum 1 risk factor for relapse.

**Figure 10 medicina-59-01224-f010:**
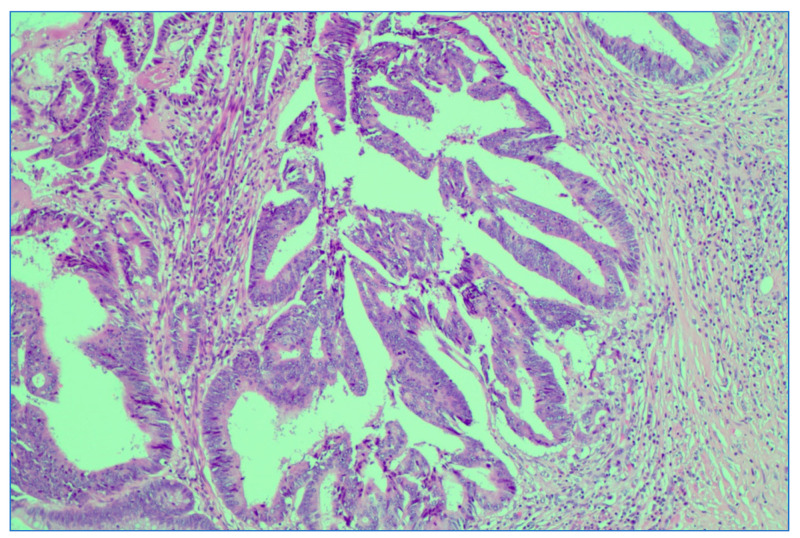
Downstage of rectal adenocarcinoma after neoadjuvant LCRCT radiochemotherapy. Illustrated is a downstaged ypT1 with invasion in submucosa; HE stain × 10 magnification.

**Figure 11 medicina-59-01224-f011:**
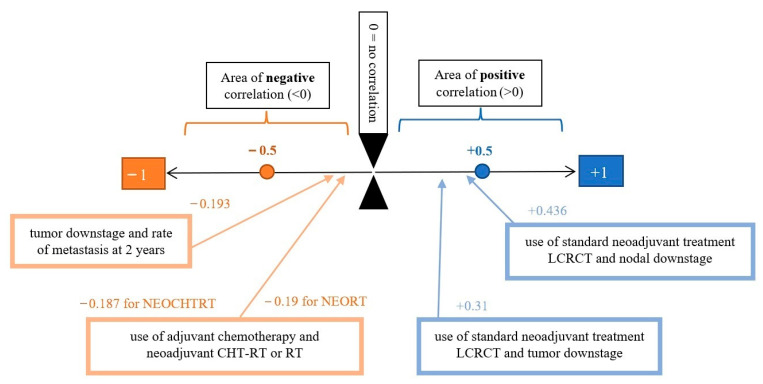
Graphical representation of the correlation between tumor downstage and metastasis rate at 2 years; adjuvant chemotherapy and neoadjuvant treatment; LCRCT and tumor downstage; and LCRCT and nodal downstage.

**Table 1 medicina-59-01224-t001:** Descriptive gender analysis.

Gender Descriptive Statistics					
		Frequency	Percent	Valid Percent	Cumulative Percent
Valid	B	99	53.2	53.2	53.2
	F	S87	46.8	46.8	100.0
	Total	186	100.0	100.0	

**Table 2 medicina-59-01224-t002:** Location of tumor formation.

LOCALIZATION					
		Frequency	Percent	Valid Percent	Cumulative Percent
Valid	JONC	45	24.2	24.2	24.2
	RI	43	23.1	23.1	47.3
	RM	59	31.7	31.7	79.0
	RS	39	21.0	21.0	100.0
	Total	186	100.0	100.0	

**Table 3 medicina-59-01224-t003:** Correlation between adjuvant chemotherapy and pathological stages T and N.

Correlations ADJCHT—pT and pN				
		ADJCHT	pT > 2	pN > 0
ADJCHT	Pearson correlation	1	0.432 **	0.462 **
	Sig. (two-tailed)		<0.001	<0.001
	N	186	186	186
pT > 2	Pearson correlation	0.432 **	1	0.120
	Sig. (two-tailed)	<0.001		0.104
	N	186	186	186
pN > 0	Pearson correlation	0.462 **	0.120	1
	Sig. (two-tailed)	<0.001	0.104	
	N	186	186	186

**—correlation is significant at the 0.01 level (2-tailed).

**Table 4 medicina-59-01224-t004:** Correlation between adjuvant chemotherapy, lymphovascular invasion, perineural invasion, resection margins, and differentiation grade.

Correlations ADJCHT—LVI, PNI, Postop, and Grade						
		ADJCHT	LVI+	PNI+	Postop+	Grade > 1
ADJCHT	Pearson correlation	1	0.135	0.188 *	0.064	0.114
	Sig. (two-tailed)		0.065	0.010	0.388	0.123
	N	186	186	186	186	186
LVI+	Pearson correlation	0.135	1	0.563 **	0.262 **	0.154 *
	Sig. (two-tailed)	0.065		<0.001	<0.001	0.036
	N	186	186	186	186	186
PNI+	Pearson correlation	0.188 *	0.563 **	1	0.236 **	0.255 **
	Sig. (two-tailed)	0.010	<0.001		0.001	<0.001
	N	186	186	186	186	186
Postop+	Pearson correlation	0.064	0.262 **	0.236 **	1	0.164 *
	Sig. (two-tailed)	0.388	<0.001	0.001		0.025
	N	186	186	186	186	186
Grade > 1	Pearson correlation	0.114	0.154 *	0.255 **	0.164 *	1
	Sig. (two-tailed)	0.123	0.036	<0.001	0.025	
	N	186	186	186	186	186

*—correlation is significant at the 0.05 level (2-tailed); **—correlation is significant at the 0.01 level (2-tailed).

**Table 5 medicina-59-01224-t005:** Correlation between adjuvant chemotherapy and minimum 1 risk factor for relapse.

Correlations ADJCHT—Min. 1 Risk Factor			
		ADJCHT	Min1RF
ADJCHT	Pearson correlation	1	0.633 **
	Sig. (two-tailed)		<0.001
	N	186	186
Min1RF	Pearson correlation	0.633 **	1
	Sig. (two-tailed)	<0.001	
	N	186	186

**—correlation is significant at the 0.01 level (2-tailed).

**Table 6 medicina-59-01224-t006:** Correlation between adjuvant chemotherapy, neoadjuvant radiochemotherapy, and neoadjuvant radiotherapy.

Correlations between ADJCHT, NEOCHTRT, and NEORT				
		ADJCHT	NEOCHTRT	NEORT
ADJCHT	Pearson correlation	1	−0.187 *	−0.190 **
	Sig. (two-tailed)		0.011	0.009
	N	186	186	186
NEOCHTRT	Pearson correlation	−0.187 *	1	0.746 **
	Sig. (two-tailed)	0.011		0.000
	N	186	186	186
NEORT	Pearson correlation	−0.190 **	0.746 **	1
	Sig. (two-tailed)	0.009	0.000	
	N	186	186	186

*—correlation is significant at the 0.05 level (2-tailed); **—correlation is significant at the 0.01 level (2-tailed).

**Table 7 medicina-59-01224-t007:** Correlation between neoadjuvant treatment, downstaging for T and N, and metastasis or local recurrence at 2 years.

Correlations between Neoadjuvant Treatment, Downstaging, and Metastasis/Relapse at 2y							
		NEORT	NEOCHTRT	T.Downstage	N.Downstage	Relapse2y	Metastasis2y
NEORT	Pearson correlation	1	0.746 **	0.338 **	0.364 **	0.100	0.110
	Sig. (two-tailed)		<0.001	<0.001	<0.001	0.174	0.133
	N	186	186	186	186	186	186
NEOCHTRT	Pearson correlation	0.746 **	1	0.310 **	0.436 **	0.131	0.082
	Sig. (two-tailed)	<0.001		<0.001	<0.001	0.074	0.269
	N	186	186	186	186	186	186
T.downstage	Pearson correlation	0.338 **	0.310 **	1	0.276 **	−0.085	−0.193 **
	Sig. (two-tailed)	<0.001	<0.001		<0.001	0.251	0.008
	N	186	186	186	186	186	186
N.downstage	Pearson correlation	0.364 **	0.436 **	0.276 **	1	−0.021	−0.102
	Sig. (two-tailed)	<0.001	<0.001	<0.001		0.777	0.165
	N	186	186	186	186	186	186
relapse2y	Pearson correlation	0.100	0.131	−0.085	−0.021	1	0.345 **
	Sig. (two-tailed)	0.174	0.074	0.251	0.777		<0.001
	N	186	186	186	186	186	186
metastasis2y	Pearson correlation	0.110	0.082	−0.193 **	−0.102	0.345 **	1
	Sig. (two-tailed)	0.133	0.269	0.008	0.165	<0.001	
	N	186	186	186	186	186	186

**—correlation is significant at the 0.01 level (2-tailed).

## Data Availability

Data available only on request due to ethical restrictions. The data presented in this study are available on request from the corresponding author and the Coltea Clinical Hospital (secretariat@coltea.ro). The data are not publicly available due to the policy of Coltea Clinical Hospital to have the approval of the Ethics Commitee for each new research study.

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
