# Peer review of "The Conditioning of Adjuvant Chemotherapy for Stage II and III Rectal Cancer Determined by Postoperative Pathological Characteristics in Romania"

_medicina, 2023, doi:10.3390/medicina59071224_

Round 1
Reviewer 1 Report
This manuscript is about using adjuvant chemotherapy in patients with locally advanced rectal cancer in Romania after the neoadjuvant radiochemotherapy and doing the Total mesorectal Excision (TME).
The authors discuss very well in their manuscript the different varieties of adjuvant chemotherapy between the American and European guidelines. And since they don't have till now their national guidelines, it's fair to mention that medical oncologists have a free choice between the NCCN and ESMO Guidelines regarding doing adjuvant chemotherapy after neoadjuvant radiochemotherapy and surgery.
The authors illustrate their data and the recently published literature well in their manuscript. Very well written with good English and with good diagrams and illustrations.
Author Response
Dear reviewer,
We want to thank you for your comments and proofing of this paper, as we want to have the highest quality of research.
With great respect.
Reviewer 2 Report
This is a single-center study from Romania reporting the preference of Romanian oncologists in prescribing adjuvant chemotherapy. The study found that Romanian oncologists preferred to apply adjuvant chemotherapy in patients with T3/4, N+, or PNI+ disease.
Major concerns:
1. Small sample size
2. Single-center study
3. The approach doesn't follow the standard guideline. It is stated that "
A positive correlation was found between ADJCHT and each of the following: (y)pT>2, (y)pN>0, and the presence of perineural invasion (PNI+)." based on the available guidelines and real-world practice, opting that a patient requires adjuvant chemotherapy (or not) is based on the clinical staging at presentation (not the post-neoadjuvant pathologic staging).
4. To highlight the importance of this study, it is stated that "There is no common ground in international guidelines on the indications of adjuvant chemotherapy (ADJCHT), with differences between the American, European and Japanese guidelines." However, this study design (with its small sample size from a single center ) cannot contribute to filling this gap. This study is a single-center, retrospective report of the LARC treatment and may not address the national approach to rectal cancer.
5. To address this question, authors need to provide a risk score (based on the impact of individual probable determining factors), run a prospective study (controlling the interpreting factors), and study the clinicians' decision on the patients' long-term outcomes. However, this study just applied correlation analysis to address this issue.
5. It is unclear whether the concomitant capecitabine dosage (852 mg/m2) is based on the specific application in Romania or is a typographical error. Based on the standard practice, the capecitabine dosage is 825 mg/m2 (five days per week) or 625 mg/m2 (daily).
Must be improved.
Author Response
Dear reviewer,
We want to thank you for your comments and proofing of this paper, as we want to have the highest quality of research. Please find attached our response to your concerns:
For points 1, 2 and 4 we would like to answer with the following clarifications:
- Our group of patients is taken from a single tertiary hospital institution, with the status of an oncological institute and with more than 20 doctors in the medical oncology specialty. The patient group was selected to meet strict inclusion criteria (e.g. capecitabine dose, radiotherapy performed with modern techniques and doses administered within the described range, surgery within 8-12 weeks, etc.). Prior to 2017 there were no potential candidate patients to enter this study, as the irradiation was done using the 2D technique. Precisely the heterogeneity given by the lack of a national guideline posed the problem of patient group selection, because not all physicians follow the same treatment principles. This is also the purpose of this study, to highlight both the preferences of Romanian oncologists in this large institution, and to warn about the need for a unified, guideline-based treatment.
- The present research is mainly conducted by the first author and the corresponding author, both PhD students. The research aims primarily to review the current situation of rectal cancer treatment in Romania. The next step of this retrospective research is to be completed with a prospective study (which is already in progress) and which will present more complete data on oncological outcome. Due to the lack of a proper period worked by us in the hospital unit, this study could only be retrospective, and its value is in our opinion essential to show what is the current situation, being practically the first Romanian study evaluating the therapeutic attitude of oncologists in this country.
- We have also mentioned these limitations of the study in the "Strenghts and limitations of the study" section at line 363
For point 3
- The difference between the NCCN, ESMO and JSSCR guidelines lies precisely in the different criteria for administering adjuvant chemotherapy. The NCCN indeed recommends the administration of adjuvant chemotherapy for all patients with LARC, but the ESMO does not. I attach 2 paragraphs taken from the latest ESMO guidelines for rectal cancer (https://doi.org/10.1093/annonc/mdx224):
“It also remains unclear whether the initial clinical (yc) or pathological (yp) stage should be used to determine the risk/benefit of adjuvant treatment. In general, downgrading in T or N stage has been recognised more as a prognostic factor of favourable outcome rather than predictive biomarker for adjuvant treatment.
Summarising, it is reasonable to consider adjuvant ChT in rectal cancer patients after preoperative CRT/RT with yp stage III (and ‘high-risk’ yp stage II). The level of scientific evidence for sufficient benefit is much lower than in colon cancer and is probably limited to DFS rather than to OS [II, C]. Hence, the decision on postoperative ChT (fluoropyrimidine alone or combined with oxaliplatin) should be risk-balanced, taking into account both the predicted toxicity for a particular patient and the risk of relapse, and should be made jointly by the individual and the clinician.”
For point 5
- We would like to mention that a prospective study is currently ongoing in the same institution and coordinated by the same authors, using as a template the PRODIGE-23 study with standard neoadjuvant treatment +/- adjuvant chemotherapy versus TNT +/- adjuvant chemotherapy. This study aims both to examine oncological outcomes and to analyse oncologists' preference to offer adjuvant chemotherapy on both arms of the study. At this time, the PhD students writing this study had limited resources available to begin research in LARC and conducted a retrospective study.
For point 6
- We have corrected the dose of capecitabine, as this was a typographical error.
We made minor corrections to the English language in the manuscript.
We hope that the explanations and changes in the manuscript will be to your liking and that you can further support the authors' scientific approach to help the development of medical research in the lagging areas such as Romania.
With great respect.
Reviewer 3 Report
There is introduction and background para ?
Retrospective study is being weakest link in the study.
Was there no marker correlation of analysis for disease ?
Any radiological correlation of disease ?
Author Response
Dear reviewer,
We want to thank you for your comments and proofing of this paper, as we want to have the highest quality of research. Please find attached our response to your concerns:
1. The introduction and background section is between lines 34 and 95 and contains a current presentation of the issues of locally advanced rectal cancer, both in terms of treatment evolution, a summary of the NCCN, ESMO and JSCCR guidelines, but also warns about the national problems present in Romania that led to the need for this study. I attach the full introduction section, in case there was a technical/computer error that did not allow you to view the introduction section of the manuscript. I will also point this out to the editor.
1.1. Overview
Locally advanced rectal cancer is most often treated with the three pillars of oncology: surgery, radiotherapy and chemotherapy with the classic order being neoadjuvant radio-chemotherapy, followed by surgery and finally by systemic therapy. Adjuvant chemotherapy (ADJCHT) after neoadjuvant treatment in rectal cancer is recommended by the American National Comprehensive Cancer Network [1], European Society for Medical Oncology [2] and Japanese Society for Cancer of the Colon and Rectum [3] guidelines on a routine basis for patients who have been treated with standard neoadjuvant treatment short course radiotherapy (SCRT) or long course radio-chemotherapy (LCRCT) for stage III. In the case of stage II rectal cancer however, there are differences between the NCCN guideline, which recommends the use of adjuvant chemotherapy systematically, versus ESMO and JSCCR guidelines, which recommend the use of adjuvant chemotherapy only in the presence of an increased risk of relapse [4].
In Romania there is no national guideline for the treatment of colorectal cancer, each oncologist having the possibility to follow international protocols and use free judgment in the decision to administer adjuvant chemotherapy. In 2021, a political initiative to promote a national cancer plan [5] has emerged, which aims to improve the management of oncological diseases. A guideline has been proposed that monitors the colorectal cancer patient pathway and recommends the use of adjuvant chemotherapy for all stage II and III rectal cancers, rather mimicking the NCCN treatment guideline, without presenting its own nationally conducted studies.
This study aims to identify the criteria chosen by Romanian oncologists to administer adjuvant chemotherapy in patients with locally-advanced rectal cancer, in the absence of a national guideline that clearly indicates this. The secondary objective is to analyze the relationships between the use of neoadjuvant treatment, the rate of tumor or nodal downstage and the rate of recurrence or metastasis at 2 years.
1.2. Background
The oncological treatment of locally advanced rectal cancer has undergone many changes in the recent decades, now using all three major oncological therapeutic options: radiotherapy, chemotherapy and surgery. However, colorectal cancer still remains the 2nd leading cause of mortality globally in 2020[6] and has an increasing incidence in Central and Eastern Europe, including Romania, where more than 4800 new cases of rectal cancer were diagnosed in 2020[7]. The appropriate administration order of the therapeutic methods was studied at the end of the 20th century, adding radiotherapy and chemotherapy to the neoadjuvant treatment scheme and establishing the total mesorectal excision (TME) technique as the gold standard surgical procedure [8,9]. The 5-year local recurrence rate ranged from 15-40%[10,11] in the 1980s, decreasing with technological advances and optimization of cancer treatment to 4-15% in the last years [12–14]. However, the reported metastasis rate (MR) for stage II-III rectal cancer is as high as 25%, with a slight improvement after the introduction of neoadjuvant radiosensitizing chemotherapy [15,16].
In recent years the possibility of total neoadjuvant treatment (TNT) including short course radiotherapy or long course radio-chemotherapy, given before or after 3-4 months of fluoropyrimidine and oxaliplatin chemotherapy, has been studied with superior results for achieving complete pathological response (pCR), better disease-free survival (DFS) and overall survival (OS)[17,18]. The RAPIDO trial [19] shows an improvement in the rate of local recurrence and metastasis rate, and the OPRA trial [20] even proposes the “watch and wait” organ preservation strategy and omitting surgery in patients with complete response [21].
The total neoadjuvant treatment strategy is not yet recommended for all stages of locally advanced rectal neoplasm (LARC)[1] and is not adopted in all centers, which is why standard neoadjuvant treatment using short-course radiotherapy or long-course radio-chemotherapy remains the basic indication in locally advanced rectal cancer.
In the case of adjuvant chemotherapy, the EORTC 22921 study indicated at first analysis a benefit of adjuvant chemotherapy for local control [22], but at the 10-year analysis it actually showed no survival benefit [23]. Another reference study is the Italian study published in 2014 showing no benefit of adjuvant fluoropyrimidine monotherapy for overall survival [24]. A large meta-analysis comprised of 4 studies concludes that adjuvant chemotherapy does not benefit overall survival, disease-free survival or metastasis rate, except for upper rectal cancers [25]. The ADORE study also adds oxaliplatin to the adjuvant treatment regimen and obtains better results for disease-free survival at 6 years (68% vs 57% HR 0.63, 95% CI 0.43-0.93), with no difference in survival [26,27].
2. The present research is mainly conducted by the first author and the corresponding author, both PhD students. The research aims primarily to review the current situation of rectal cancer treatment in Romania. The next step of this retrospective research is to be completed with a prospective study (which is already in progress) and which will present more complete data on oncological outcome. Due to the lack of a proper period worked by us in the hospital unit, this study could only be retrospective, and its value is in our opinion essential to show what is the current situation, being practically the first Romanian study evaluating the therapeutic attitude of oncologists in this country. We have also mentioned these limitations of the study in the "Strenghts and limitations of the study" section at line 363
3. The main objective of the study is to analyse the relationship between disease markers (LARC with specific histopathological features: (y)pT, (y)pN, LVI, PNI, Postop and Grade ) and the use or not of adjuvant therapy. The analysis uses Pearson correlations and T-tests using IBM SPSS v29.
4. Radiological information was used only for the secondary objective study, specifically for tumour and nodal downstage analysis. Their relation with metastasis or local control at 2 years was performed. The study focused mainly on histopathological information and this was highlighted throughout the study, including the addition of specific figures for the findings in the study (Fig 6, 7, 8).
We hope that the explanations and changes in the manuscript will be to your liking and that you can further support the authors' scientific approach to help the development of medical research in the lagging areas such as Romania.
With great respect.
Round 2
Reviewer 2 Report
Thanks to the authors for the clarifications and responses to the comments. However, the submitted manuscript doesn't provide enough educational messages to pass the review process. It is suggested the manuscript be submitted to the national journals to guide Romanian clinicians better.
The manuscript still requires minor English copy editing.